# Comparative efficacy of schroth and core training for early postoperative recovery in adolescent idiopathic scoliosis: A single blind randomized controlled trial

Fanyuan Meng[1☉], Kerong Li[1,2☉], Wei Wang[3], Rui Yang[1], Cong Wang[1], Zhi Zhao[4], Moxian Chen[1]*, Lijuan Ao[1]*

1 School of Rehabilitation, Kunming Medical University, Kunming, Yunnan, China, 2 Department of Rehabilitation, Kunming Municipal Hospital of Traditional Chinese Medicine, Kunming, Yunnan, China, 3 College of Mechanical and Electrical Engineering, Harbin Engineering University, Harbin, Heilongjiang, China, 4 Department of Orthopaedics, The Second Affiliated Hospital of Kunming Medical University, Kunming, China

☉ These authors contributed equally to the work and should be regarded as co-first authors.
* aolijuan@kmmu.edu.cn (LA); chenmoxian@kmmu.edu.cn (MC)

## Abstract

### Background

To evaluate the clinical efficacy of Schroth exercises combined with core training versus core training alone on early trunk balance optimization and functional recovery in adolescent idiopathic scoliosis (AIS) patients following selective thoracic fusion surgery.

### Methods

This single-blinded randomized controlled trial enrolled 46 AIS patients with Lenke1 type configuration who underwent selective thoracic posterior spinal fusion at the Orthopedics Department of the Second Affiliated Hospital of Kunming Medical University. Participants were randomly assigned to either the Schroth exercise combined Core Training (SCT, n = 23) or Core training group (CT, n = 23). The SCT group received Schroth three-dimensional (3D) scoliosis-specific exercises combined with core stabilization training: during the initial 3 postoperative months, they performed core exercises, rotational angular breathing, and daily postural management; during the subsequent 3 months, they engaged in Schroth 3D corrective exercises targeting surgical correction outcomes. The CT group exclusively performed core stabilization training throughout the 6-month postoperative period. Both groups received 40-minute intervention sessions three to four times weekly from discharge to 6 months postoperatively. Assessments were conducted at four time points: preoperatively, at the discharge (postoperative day 7), and at 3 and 6 months postoperatively.

**Data availability statement:** The minimal dataset required to replicate the findings of this study has been deposited in Figshare and is available at https://doi.org/10.6084/m9.figshare.30836405.

**Funding:** This work was supported by the Yunnan Provincial Department of Science and Technology-Kunming Medical University Joint Special Project for Applied Basic Research (202201AY070001-014), 2024 Yunnan Provincial University Science and Technology Project for Serving Key Industries (FWCY-BSPY2024074), Natural Science Foundation of Guangdong Province (2022A1515010169). The funders had no role in study design, data collection and analysis, decision to publish, or preparation of the manuscript.

**Competing interests:** The authors have declared that no competing interests exist.

Spinal radiographic parameters, paraspinal muscle surface electromyography, core muscle strength and endurance, and SRS-22 questionnaire data were analyzed to evaluate early clinical efficacy across three domains: body structure/function, activities, and participation in surgically treated AIS patients.

## Results

Analysis of the primary radiographic outcomes revealed no significant time-by-group interactions. However, the SCT group demonstrated superior overall improvement in pelvic balance compared to the CT group (Group main effect: $p = 0.032$). For secondary exploratory outcomes, significant interactions were observed for trunk extensor endurance and SRS-22 self-image ($p < 0.01$), with the SCT group showing greater improvement than the CT group specifically at the 6-month follow-up ($p < 0.05$). In unadjusted exploratory comparisons, trunk flexor endurance was also better in the SCT group at 6 months ($p = 0.046$). No other significant between-group differences were found.

## Conclusion

Compared with isolated core stabilization training, Schroth 3D scoliosis-specific exercises combined with core training demonstrate superior efficacy in improving early postoperative muscular function, pelvic symmetry, and self-image. Nevertheless, comparable effects were observed between both rehabilitation protocols regarding early postoperative Cobb angles of major and minor curves, convex-concave paraspinal muscle balance restoration, pain alleviation, and psychological status improvement.

## Introduction

Adolescent idiopathic scoliosis (AIS) is a three-dimensional (3D) spinal deformity of undetermined etiology, diagnosed when Cobb angle measures ≥10° and secondary causes have been excluded [1,2]. The International Society on Scoliosis Orthopaedic and Rehabilitation Treatment (SOSORT) 2016 guidelines [3] recommend management options including observation, conservative treatment, or surgery. Conservative treatments currently available include bracing therapy and physiotherapy, which effectively control progression in approximately 90% of patients [3]. However, surgical intervention was warranted when the Cobb angle exceeds 45°-50°. The primary goals of surgical intervention are to halt scoliosis progression, correct the deformity, and restore coronal-sagittal spinal balance [4]. The main procedures for AIS are anterior spinal fusion (ASF) and posterior spinal fusion (PSF). PSF has emerged as the predominant procedure due to its superior ability to provide greater corrective strength and enhance stability across all three spine columns [5]. In certain scoliosis patients, a dominant main thoracic (MT) curve is accompanied by a compensatory lumbar (L) or thoracolumbar (TL) curve. If the lumbar curve demonstrated good flexibility (correcting to ≤25° on side-bending radiographs or meeting non-structural curve

criteria), along with mild apical vertebral translation (AVT) and rotation (Nash-Moe grade), and no significant trunk imbalance, PSF with selective thoracic fusion may be considered [6]. Surgeons selectively fused the major curve while preserving the minor curve to maintain spinal flexibility and optimize quality of life. However, scoliosis-specific physical therapy targeting unfused secondary curves remains essential to prevent their postoperative progression.

Numerous clinical studies have found pain, compromised pulmonary function, and postural control deficits in patients after PSF [4]. Long-term studies found that even 20 years post-surgery, AIS patients continue to demonstrate inferior spinal core musculature endurance compared to healthy peers [7]. Cetinkaya et al. [8] demonstrated that post-surgical movement limitations all planes adversely affected motor function, quality of life, and physical appearance. Spine convex-concave paraspinal muscular imbalance existed in AIS, with the apical vertebral regions showing the most pronounced manifestations [9]. Fusion surgery improves paraspinal muscles asymmetry in AIS but did not restore it to normal subjects level [10,11]. The meta-analysis by Wang et al. reported a 9.2% incidence of adding-on phenomenon after selective thoracic fusion, primarily associated with skeletal immaturity (Risser ≤2), major curves exceeding 60°, and residual thoracic curves greater than 20° postoperatively [12].

The SOSORT guidelines [3] recommend postoperative physical therapy (Grade B evidence, Class II recommendation) with the goals of preventing progression of unfused curves, enhancing pulmonary function, and improving quality of life in patients after spinal fusion. Current consensus indicates the 6-to-12-week postoperative period as the critical soft-tissue healing phase, during which patients may initiate early mobilization and engage in non-impact activities [13]. In the early postoperative period (within 12 weeks), core stability training may be considered with caution due to safety concerns. While such training lacks specificity for addressing unfused secondary curves, it can facilitate early muscular recovery following surgery. Contemporary spinal constructs demonstrate adequate load-bearing capacity to support pre-fusion rehabilitation, permitting the implementation of scoliosis-specific training protocols beyond 12 weeks postoperatively, guided by radiographic evaluation. Research examining AIS patients post-surgical correction found that implementing five basic Schroth 3D corrective exercises (30 min daily for 3–6 months, beginning 2–3 weeks postoperatively) maintained their curves between 22–33° over a follow-up period of 6 months to 2 years [14]. Despite theoretical benefits, the application of Schroth exercises in postoperative scoliosis management lacks support from Randomized controlled trials (RCT), underscoring a critical research gap. This study employed a single-blind RCT design to compare the effectiveness of Schroth exercises with core stabilization versus core training alone on maintaining postural correction and restoring spinal alignment in AIS following selective thoracic fusion. Secondary objectives included assessing the intervention's impact on trunk muscle functional recovery and enhancing the quality of life. We hypothesized that at the six-month follow-up, patients receiving Schroth exercises combined with core stabilization would demonstrate significantly greater improvement in the maintenance of postural correction and spinal alignment than those receiving core training alone.

## Methods

### Trial design

This study was a single-blind randomized controlled study. The study was conducted in accordance with the Declaration of Helsinki, approved by the Chinese Clinical Trial Registry, registered at http://www.chictr.org.cn/ (ChiCTR2300070979) and approved by the Ethics Committee of Kunming Medical University (No. KMMU2022MEC093). All patients and their parents or guardians were informed of the purpose and protocol of this study before participation. Before the random grouping was carried out, the patients and their parents or guardians had already signed the informed consent form.

### Participants

Subjects were from AIS patients attending the Department of Orthopaedics of the Second Affiliated Hospital of Kunming Medical University. Participant recruitment for this study commenced on July 17, 2023 and was completed on September 10, 2024. Forty-six patients were included according to the inclusion and exclusion criteria.

**Inclusion criteria.**

(1) Female patients with radiologically confirmed AIS;

(2) Aged 10–18 years;

(3) Preoperative Cobb angle >40° and scheduled for PSF;

(4) Scoliosis classification: Lenke type 1;

(5) Scheduled for thoracic spinal fusion only;

(6) Voluntary participation with signed written informed consent.

**Exclusion criteria.**

(1) patients with growth and developmental disorders, history of neurological, musculoskeletal and skeletal infections, etc;

(2) patients with cognitive impairments that prevent them from cooperating with exercise training and completing the relevant questionnaires;

(3) patients with a history of other major surgeries;

(4) patients who refuse or are unwilling to participate in the study.

Participants were randomized into two groups using random number table methodology: Schroth exercise combined Core Training (SCT) and Core training group (CT). Subjects remained blinded to their grouping throughout the study. In this study, a 6-month physiotherapy intervention was conducted in both groups from hospital discharge to 6 months postoperatively. Different training methods were developed for the two groups according to the acute and subacute postoperative period (first 3 months postoperatively) and the recovery period (months 4–6). The SCT protocol consisted of Schroth 3D scoliosis-specific corrective exercises combined with core stabilization training. The Schroth methodology incorporated 3D plane corrective maneuvers tailored to postoperative correction outcomes (particularly targeting the minor curve), rotational angular breathing exercises, and activities of daily living (ADL) postural management education [15]. The CT intervention comprised core stabilization training alone [16]. Initially, 23 subjects were enrolled in each group. During the study period, data were missing because subjects failed to undergo complete postoperative physiotherapy (SCT: 2, CT: 3) and withdrew without postoperative follow-up (SCT: 3, CT: 2). The final analysis included 36 subjects (18 in each group) (Fig 1).

## Intervention

In line with Enhanced Recovery After Surgery (ERAS) principles and following a safety evaluation conducted in collaboration with the surgeon, the rehabilitation protocol commenced on the first postoperative day. The program adhered to a principle of gradual progression. It began with 4–5 initial outpatient sessions conducted by the same certified Schroth physical therapist (PT) to ensure that participants and their caregivers could accurately perform the exercises independently at home or school. Between the weekly outpatient sessions, both groups were required to perform the prescribed exercises at home to consolidate learning and maintain continuity of care. Throughout the study period, weekly outpatient sessions with the same PT continued, allowing the therapist to monitor performance and adjust the home-based training program based on individual competency and the quality of exercise completion.

Following the principle of gradual and periodic exercise training, the SCT group began with low-intensity training during the first 3 months. The protocol encompassed core muscle training, rotational angular breathing, and ADL posture management. During postoperative months 4–6, Schroth 3D scoliosis-specific corrective exercises were administered. These exercises were tailored to postoperative correction status, incorporating three-plane corrective maneuvers, rotational

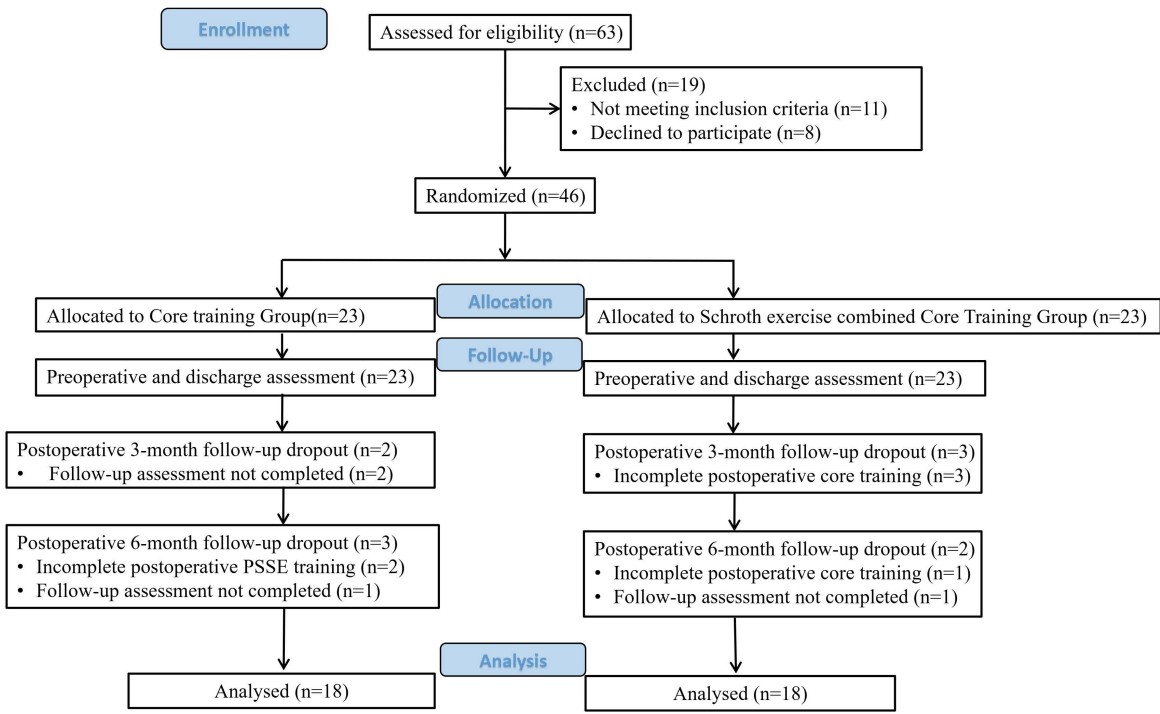

**Fig 1. The CONSORT diagram of the study.**

angular breathing techniques, and ADL posture management. Each patient received an individualized Schroth protocol with 2–3 specific 3D corrective maneuvers based on their surgical outcomes by the same certified Schroth PT (Fig 2). ADL posture management covered sitting, standing, walking, and sleeping postures. The CT group protocol consisted exclusively of core stabilization training (Fig 3). Both groups performed 40-minute sessions 3–4 times weekly. To ensure safety and efficacy compliance, systematic follow-up were implemented for all participants: weekly telephonic/WeChat communication with parents, mandatory video recording submission for therapist validation per session, progress tracking, real-time feedback adjustments, and outpatient retraining/supervision when clinically indicated.

## Outcomes

### Primary outcomes

**Imaging indicators.** Participants underwent standing whole-spine anteroposterior and lateral radiographs at preoperative, discharge, 3-month postoperative, and 6-month postoperative follow-ups. At each predefined assessment timepoint, radiographic parameters (including Cobb angles) were measured from standing radiographs using RadiAnt DICOM Viewer software. (1) Cobb angle: The upper end vertebrae (UEV) and lower end vertebrae (LEV) of the primary curve were identified on radiographs. Tangent lines to the superior endplate of UEV and inferior endplate of LEV were constructed. The angle between these parallel lines defined the Cobb angle. The apical vertebra (AV) was the vertebral body demonstrating maximal deviation from the scoliotic midline and maximal horizontal plane rotation [17] (Fig 4A). (2) Shoulder balance: the height difference between the two shoulders was evaluated by measuring the soft tissue image above the acromioclavicular joints in the coronal plane [18] (Fig 4C). (3) Pelvic balance: The height difference between the highest points of bilateral iliac crests was measured in coronal projection [19] (Fig 4D). (4) Coronal balance: The center sacral vertical line (CSVL) was defined as a

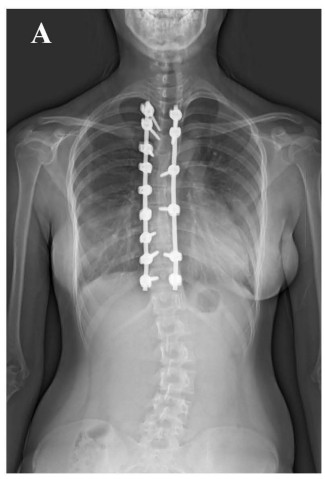
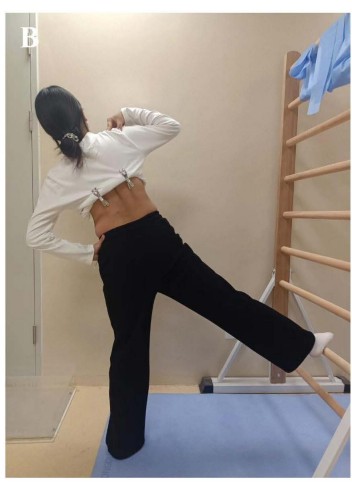
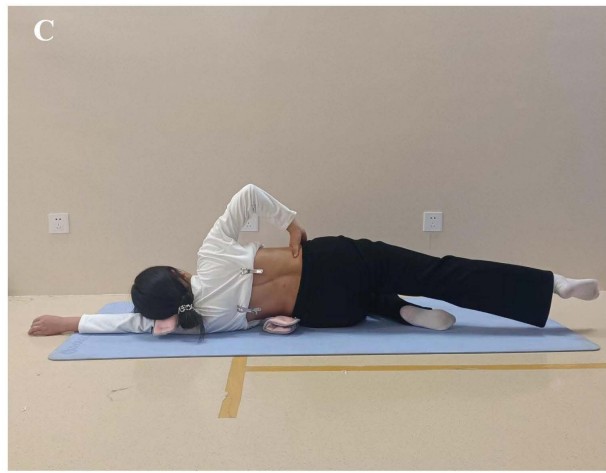

**Fig 2. Condition-specific physiotherapy for thoracically fused patients. (A)** Postoperative radiographs of thoracically fused patients. **(B)** Muscle cylinder maneuver. **(C)** Lateral recumbent exercises.

vertical line passing through the midpoint of S1 superior endplate. The cervical 7 plumb line (C7PL) was a vertical line descending from the midpoint of C7 vertebral body. The position of the C7PL in relation to the CSVL defined the presence or absence of coronal plane imbalance [20] (Fig 4B).

### Secondary outcomes

**Paraspinal muscle surface electromyography (sEMG).** sEMG data from paraspinal musculature were collected using a sEMG monitoring system (NORAXON, USA) in a quiet, comfortable environment, focusing on the UEV, LEV, and AV regions of the primary curve. (1) Electrode placement: After exposing the patient's back, the skin surrounding the spine was cleansed with 75% ethanol-soaked gauze to remove oils and dust, ensuring optimal skin-electrode interface. Electrode sheets were positioned bilaterally on the erector spinae muscles, 2 cm lateral to UVE, LVE and AV in the thoracic segment. Sensors were attached 1 cm next to the electrode sheet, connecting the sensors and electrode sheet, with a total of 3 pairs (6 sensors) [21] (Fig 5). (2) Testing protocol: Three standardized actions were performed: Action 1: Static standing maintained for 30 s; Action 2: ABT testing posture held for 30 s; Action 3: BST testing posture sustained for 30 s; A 5 min intervals separated each action. Root mean square (RMS) values were calculated to reflect motor units recruitment, activation patterns, muscular coordination, and synchronization during muscle activity [22]. Paraspinal muscle symmetry index (PMSI) quantified bilateral sEMG symmetry, calculated as PMSI = convex-side RMS/concave-side RMS. Values approaching 1 indicate superior symmetry, while deviations from 1 denote asymmetry [23].

**Evaluation of spinal flexor and extensor endurance.** The Biering-Sorensen test (BST) was performed with the participant positioned prone on the treatment plinth. The pelvis and lower extremities were immobilized while the upper body overhung the plinth edge, maintaining strict horizontal alignment. Participants crossed their arms over the chest with paraspinal musculature isometrically resisting gravitational forces equivalent to body mass superior to the anterior superior iliac spine. Testing ceased when the upper body failed to maintain horizontal positioning [24] (Fig 6A). For assessment of abdominal muscle endurance, the abdominal test (ABT) required participants to sit on the treatment plinth with the trunk maintained at a 60° angle relative to the bed surface. Knees and hips were flexed at 90°, arms crossed over the chest. Participants sustained this posture until trunk inclination fell below 60°, terminating the assessment [25] (Fig 6B).

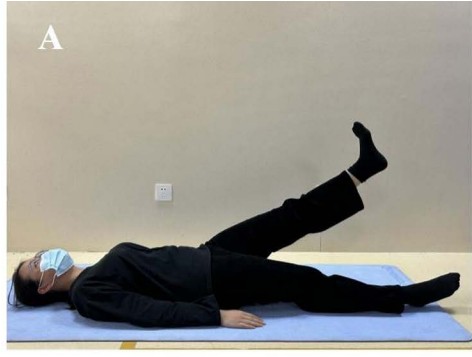
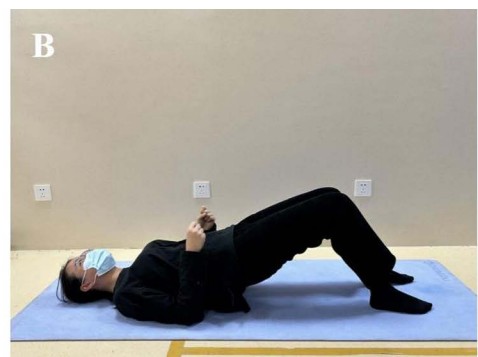
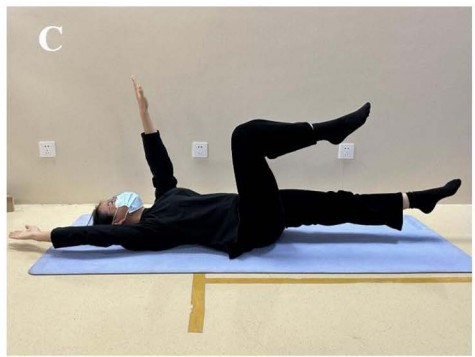
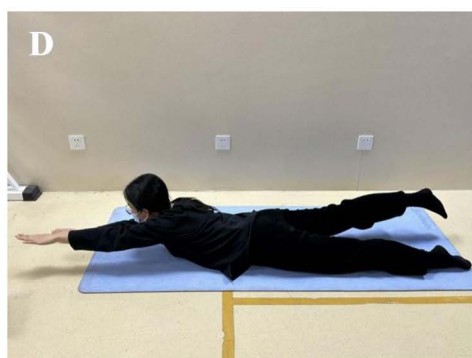
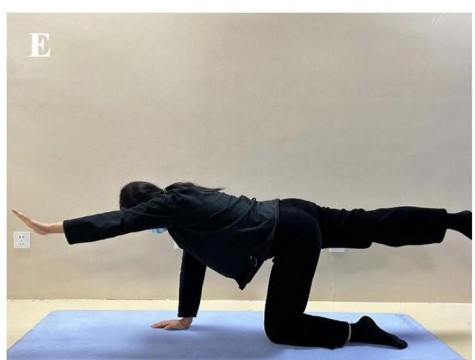

**Fig 3. Core stabilization training. (A)** Single-leg straight leg raise and lower. **(B)** Glute bridge. **(C)** Dead bug exercise. **(D)** Unilateral superman lift. **(E)** Kneeling back extension.

**Spinal flexor and extensor strength testing.** During the standardized ABT and BST test positions, muscle strength assessment was conducted using a portable plyometric instrument (JTECH Medical, USA). The evaluator applied opposing forces against which patient generated maximal isometric contractions of the trunk flexors and extensors, respectively, maintaining each contraction 3 s. Repeat the test 3 times, rest for 30 s each time. The average value was calculated for analysis (Fig 7) [26].

**SRS-22.** The SRS-22 questionnaire serves as a valid and essential instrument for evaluating quality of life in scoliosis patients, extensively utilized during preoperative assessment and postoperative follow-up. Completed by participants in a

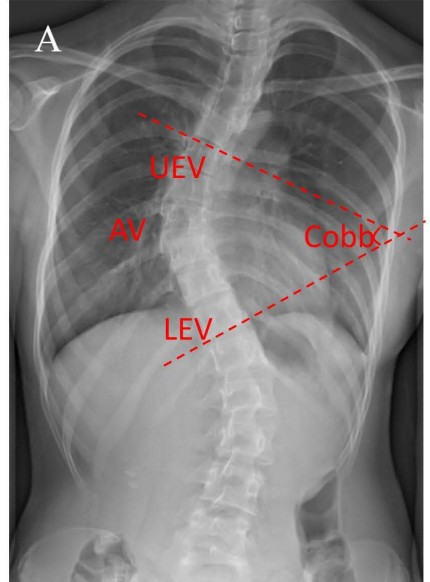
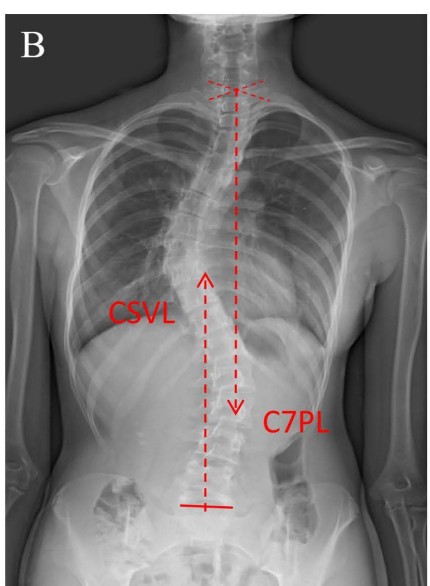

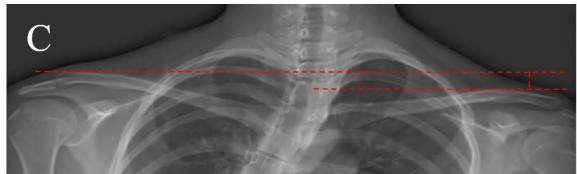
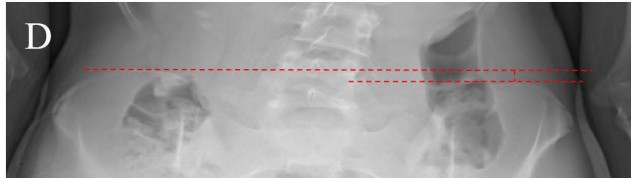

**Fig 4. Imaging indicators. (A)** The measurement methodology of the Cobb angle on the anteroposterior radiograph. **(B)** The measurement methodology of the CSVL for coronal balance. **(C)** The measurement methodology of shoulder balance. **(D)** The measurement methodology of pelvic balance.

quiet environment, this instrument comprises 22 items assessing five domains: pain, functional status, self-image, mental health, and satisfaction with treatment. Each item assigns scores ranging from 1 (lowest) to 5 (highest). Domain scores represent arithmetic means of corresponding item responses. The SRS-22 demonstrates high reliability and validity, with higher scores indicating less impact of scoliosis on patients' quality of life. This enables comprehensive evaluation of subjective perceptions regarding disease status and clinical management efficacy, thereby quantifying AIS quality of life [27].

## Sample size

The sample size of this test was calculated based on the statistical software G-Power software 3.1.9.7 (http://www.gpower.hhu.de). Based on previously reported effect size ($f = 0.25$) from Hikmet et al. [28] the sample size was determined using the following parameters: $\alpha = 0.05$ (two-sided test), and statistical power $(1-\beta) = 0.90$. The final sample content was calculated to be $n = 36$. To account for an anticipated 20% attrition rate, the final target enrollment was set at 44 participants (22 per group) to ensure adequate statistical power sensitivity.

## Randomization and blinding

Forty-six AIS patients were randomly divided into two groups: SCT group and CT group. Matched pairs randomization was performed using the Research Randomizer program (randomizer.org) with patients matching criteria

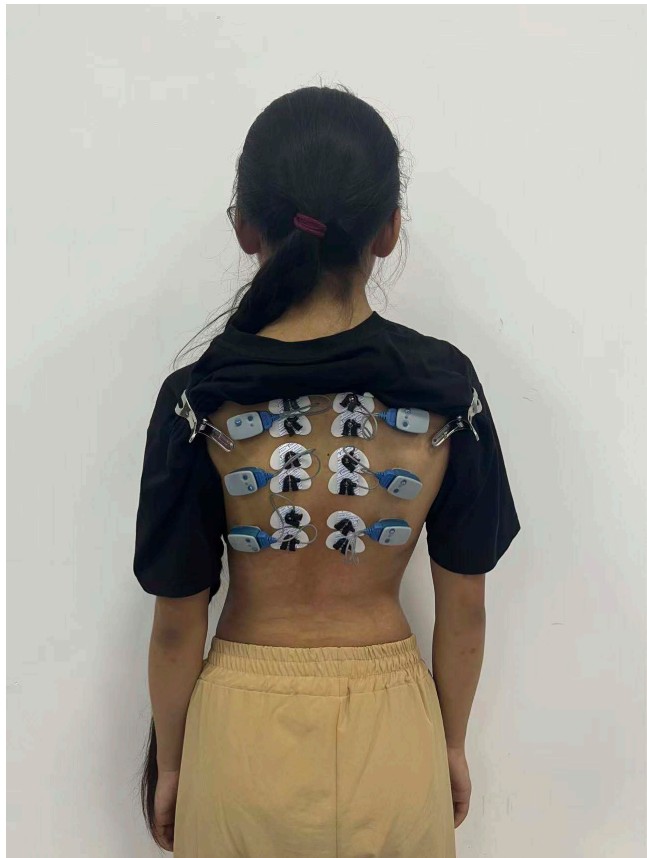

**Fig 5. Paraspinal muscle surface electromyography of major curve.**

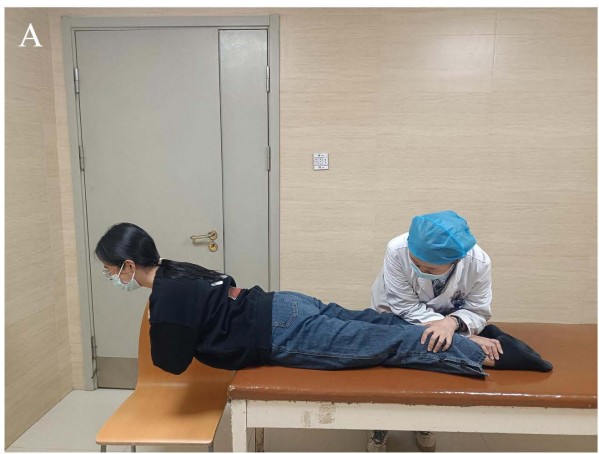 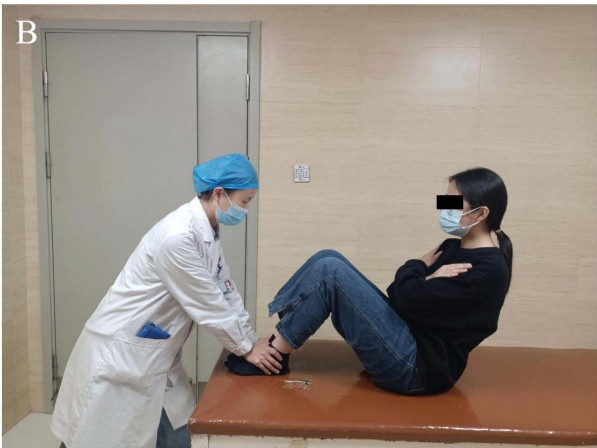

**Fig 6. Spinal flexor and extensor endurance testing. (A)** The Biering-Sorensen testing position. **(B)** The abdominal muscle testing position.

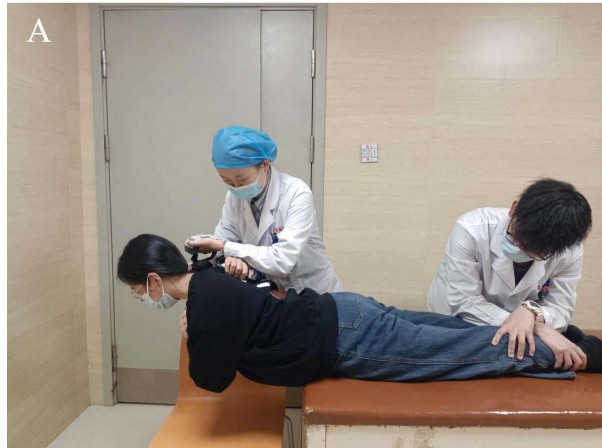
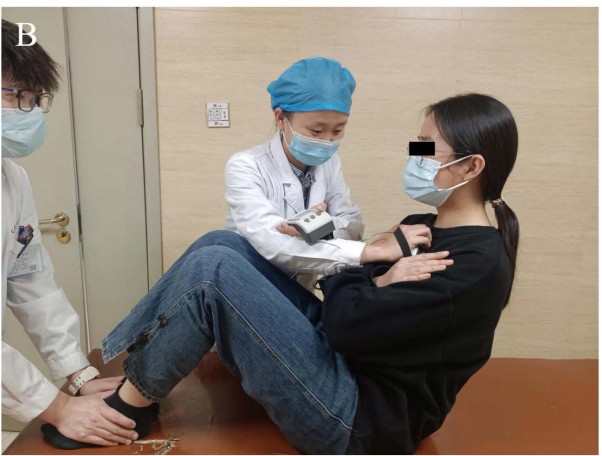

**Fig 7. Spinal flexor and extensor strength testing. (A)** Spinal extensor muscle strength assessment. **(B)** Spinal flexor muscle strength assessment.

including Cobb angle, age, body mass index (BMI), and Risser sign. This study employed a single-blind method for patients. The informed consent was obtained from all participants and their guardians prior to randomization. Participants were informed that they would be randomly assigned to one of two exercise regimens, but were blinded to their specific group allocation throughout the trial. All procedures were approved by the institutional ethics committee. All assessments were evaluated by the same physiotherapist (PT) at baseline, discharge, 3 months, and 6 months post-operation.

## Statistical methods

Continuous variables underwent normality testing via the Shapiro-Wilk (S-W) test. Normally distributed data were presented as mean ± standard deviation (SCT). Categorical variables are summarized as frequencies and percentages. Baseline comparisons (SCT vs CT) utilized independent-samples t-tests for normally distributed data, and chi-square test for categorical variables, as appropriate. For non-normal distribution data, logarithmic transformation is applied and then two-factor repeated measures analysis of variance is conducted. A two-way repeated measures ANOVA was used to assess interaction and main effects of time and group. If the assumption of sphericity was violated, the Greenhouse-Geisser correction was applied. In the case of a significant interaction or main effect, post-hoc analyses were conducted with Bonferroni correction for multiple comparisons. All statistical tests were two-tailed, and a p-value < 0.05 was considered statistically significant for primary analyses. Analyses of secondary outcomes were considered exploratory, and no adjustments for multiple comparisons were made for these analyses. All statistical analyses were performed using IBM SPSS Statistics version 25.0.

## Results

### Characteristics of the study population

No severe adverse events (including instrumentation fracture, rod breakage, or reoperation) occurred throughout the trial period. All participants reported no severe pain during exercise interventions. Table 1 shows the demographic information of the participants. The two groups demonstrated no statistically significant differences in preoperative baseline characteristics (P > 0.05). Similarly, no statistically significant intergroup differences existed in AIS-related demographic and clinical features (P > 0.05) (Table 1).

**Table 1. Baseline clinical and demographic characteristics of groups. [n (%)/ Mean ± SD].**

| | | SCT (n = 18) | CT (n = 18) | X²/ T/ Z | P |
|---|---|---|---|---|---|
| Sex (female) | | 18 (100%) | 18 (100%) | | |
| Age (years) | | 14.50 ± 1.88 | 14.05 ± 1.73 | 0.736 | 0.467 |
| Arm length (cm) | | 158.83 ± 6.03 | 158.22 ± 6.89 | 0.283 | 0.779 |
| Height (cm) | | 154.67 ± 5.54 | 147.48 ± 36.83 | 0.796 | 0.431 |
| Body weight (kg) | | 44.50 ± 7.30 | 42.38 ± 5.88 | 0.955 | 0.346 |
| Body mass index | | 17.59 ± 2.33 | 16.91 ± 1.93 | 0.955 | 0.354 |
| Risser sign (°) | | | | | |
| | 1 | 0 | 0 | 3.038 | 0.401 |
| | 2 | 3 (16%) | 2 (11%) | | |
| | 3 | 3 (16%) | 7 (39%) | | |
| | 4 | 7 (39%) | 7 (39%) | | |
| | 5 | 5 (29%) | 2 (11%) | | |
| Curve type | | | | | |
| | Lenke1A | 9 (50.0%) | 10 (55.6%) | 0.586 | 0.746 |
| | Lenke1B | 7 (38.9%) | 5 (27.8%) | | |
| | Lenke1C | 2 (11.1%) | 3 (16.7%) | | |
| Major-curve ATR (°) | | 16.05 ± 4.27 | 15.00 ± 5.30 | 0.657 | 0.515 |
| Major-curve Cobb (°) | | 52.44 ± 11.79 | 52.94 ± 13.92 | −0.116 | 0.908 |
| Minor-curve Cobb (°) | | 30.83 ± 11.17 | 29.88 ± 10.35 | 0.263 | 0.794 |
| The number of fused vertebrae | | 8.05 ± 0.87 | 7.77 ± 0.73 | 1.035 | 0.308 |
| Postoperative major-curve Cobb (°) | | 15.88 ± 7.37 | 16.55 ± 7.64 | −0.266 | 0.792 |
| Postoperative minor-curve Cobb (°) | | 10.27 ± 6.01 | 12.05 ± 9.39 | −0.676 | 0.504 |
| Major-curve correction rate (%) | | 70.44 ± 9.94 | 67.61 ± 14.30 | 0.690 | 0.495 |
| Minor-curve correction rate (%) | | 66.00 ± 15.33 | 62.50 ± 24.68 | 0.511 | 0.613 |

## Comparison of radiographic postural correction outcomes between the two groups

The two-way repeated-measures ANOVA revealed no significant interaction effects for any imaging outcomes (P > 0.05). For pelvic balance, significant main effects were found for both Group ($F_{group}$ = 4.070, $P_{group}$ = 0.032) and Time ($F_{time}$ = 4.683, $P_{time}$ = 0.038), with the SCT group demonstrating superior correction compared to the CT group (p < 0.05). The main effects for the other four indicators were not significant. Detailed results are provided in Table 2.

## Comparison of convex/concave RMS values in paraspinal muscles between the two groups

sEMG assessment was not performed at postoperative discharge due to surgical wound considerations. The two-way repeated-measures ANOVA detected no significant Time × Group interaction for any sEMG measure (Table 3). During static standing, flexor testing, and extensor testing performed preoperatively and at 3-month and 6-month postoperative intervals, no statistically significant differences were observed between the SCT and CT groups in RMS values of paraspinal musculature at the AV, UEV, and LEV of the major curve (P > 0.05). At both 3 months and 6 months postoperatively, convex/concave RMS values of AV paraspinal muscles in both groups approached closer to 1 compared to preoperative baselines (P < 0.05), indicating enhanced major-curve paraspinal muscles symmetry resulting from surgical intervention. Complete results are presented in Table 4.

**Table 2. Comparative analysis of postural optimization outcomes between the two patient cohorts.**

| Outcomes | | T1 Mean±SD | T2 Mean±SD | T3 Mean±SD | $F_{group}$, $P_{group}$ | $F_{time}$, $P_{time}$ | Time*Group | | | Pairwise comparison |
|---|---|---|---|---|---|---|---|---|---|---|
| | | | | | | | F | P | Partial $\eta^2$ | |
| Major-curve Cobb (°) | SCT | 15.88±7.37 | 15.38±7.64 | 15.72±7.94 | 0.640, 0.487 | 0.026, 0.873 | 0.426 | 0.601 | 0.012 | None |
| | CT | 16.55±7.64 | 16.55±7.58 | 16.38±7.87 | | | | | | |
| Minor-curve Cobb (°) | SCT | 10.27±6.01 | 12.05±9.39 | 11.94±5.68 | 0.674, 0.482 | 0.277, 0.602 | 0.416 | 0.661 | 0.016 | None |
| | CT | 12.05±9.39 | 11.44±8.36 | 12.44±8.17 | | | | | | |
| Shoulder balance (cm) | SCT | 0.60±0.58 | 0.62±0.55 | 0.50±0.48 | 0.003, 0.990 | 2.732, 0.108 | 0.676 | 0.480 | 0.019 | None |
| | CT | 0.78±0.74 | 0.67±0.43 | 0.78±0.46 | | | | | | |
| Pelvic balance (cm) | SCT | 0.32±0.39 | 0.39±0.35 | 0.39±0.42 | 4.070, 0.032* | 4.683, 0.038* | 2.211 | 0.131 | 0.031 | None |
| | CT | 0.40±0.43 | 0.55±0.35 | 0.80±0.48 | | | | | | |
| Coronal balance (cm) | SCT | 1.08±0.90 | 1.01±0.87 | 0.92±0.67 | 0.549, 0.582 | 0.448, 0.508 | 0.403 | 0.607 | 0.012 | None |
| | CT | 0.90±0.63 | 0.75±0.65 | 0.91±0.67 | | | | | | |

T1: Discharge; T2: 3-month postoperative; T3: 6-month postoperative; *: $P < 0.05$; **: $P < 0.01$.

## Comparative analysis of changes in trunk muscular strength and endurance between the two groups

Assessment of trunk muscular strength and endurance was not performed at postoperative discharge due to wound pain concerns and potential instrumentation loosening from excessive movement. A two-way repeated-measures ANOVA revealed a significant Group × Time interaction for trunk extensor endurance (F = 6.188, P = 0.030). Simple effect analysis showed the SCT group demonstrated superior extensor endurance to the CT group specifically at the 6-month assessment (F = 4.817, P = 0.035), with no differences at baseline or 3 months. No other significant interactions or main effects were found for the remaining strength and endurance measures (P > 0.05; see Table 3). As an exploratory analysis of this secondary endpoint, an unadjusted between-group comparison at 6 months suggested the SCT group also had greater trunk flexor endurance (83.61 ± 26.22 vs. 66.77 ± 22.47, p = .046). At the 3-month postoperative assessment, both groups exhibited significantly reduced flexor and extensor strength versus preoperative baselines (P < 0.01). The SCT group demonstrated significantly diminished extensor endurance compared to preoperative values (P < 0.01), while the CT group showed significantly decreased endurance in both flexor and extensor musculature (P < 0.01). By the 6-months postoperative assessment, flexor/extensor strength and endurance in the SCT group recovered to preoperative levels (P > 0.05). The CT group exhibited recovery of flexor endurance to preoperative status (P > 0.05), but flexor/extensor strength and extensor endurance remained significantly inferior to preoperative baselines (P < 0.01). No significant intergroup differences were observed in trunk flexor/extensor strength or endurance at 3 months postoperatively (P > 0.05). Detailed results regarding flexor/extensor strength or endurance are presented in Table 5 and Fig 8.

## Comparison of SRS-22 scores between the two groups

A two-way repeated-measures ANOVA for the SRS-22 revealed a significant Group × Time interaction for self-image (F = 4.105, P = 0.009). Simple effect analysis showed that while self-image scores did not differ between groups preoperatively or at 3 months, the SCT group reported significantly higher scores than the CT group at 6 months (F = 4.492, P = 0.041). Within the SCT group, post-hoc tests confirmed that self-image improved significantly from baseline to both 3 and 6 months (P < 0.001), and further improved from discharge to 6 months (P = 0.003). No significant main or interaction effects were found for the other SRS-22 subscales in Table 6. As a secondary outcome measure in the exploratory analysis, no statistically significant differences were observed between the two groups for any domain scores preoperatively, at discharge, and at the 3-month postoperative interval (P > 0.05).

**Table 3. Comparative analysis of convex/concave RMS outcomes and trunk muscular strength and endurance in paraspinal muscles between the two patient cohorts across three testing maneuvers within the groups.**

| Outcomes | | | T0 Mean±SD | T2 Mean±SD | T3 Mean±SD | Time*Group F | P | Partial η² | Pairwise comparison |
|---|---|---|---|---|---|---|---|---|---|
| Static standing Convex/ Concave RMS | UEV | SCT | 1.33±0.31 | 1.20±0.33 | 1.15±0.41 | 0.409 | 0.648 | 0.012 | None |
| | | CT | 1.25±0.34 | 1.19±0.26 | 1.18±0.34 | | | | |
| | AV | SCT | 1.78±0.53 | 1.20±0.26 | 1.19±0.31 | 0.058 | 0.944 | 0.002 | None |
| | | CT | 1.61±0.49 | 1.18±0.34 | 1.20±0.35 | | | | |
| | LEV | SCT | 1.48±0.40 | 1.15±0.28 | 1.13±0.39 | 0.656 | 0.503 | 0.019 | None |
| | | CT | 1.35±0.32 | 1.18±0.16 | 1.14±0.29 | | | | |
| Flexor testing Convex/ Concave RMS | UEV | SCT | 1.48±0.39 | 1.13±0.27 | 1.21±0.33 | 0.508 | 0.587 | 0.015 | None |
| | | CT | 1.33±0.32 | 1.12±0.17 | 1.14±0.20 | | | | |
| | AV | SCT | 1.56±0.73 | 1.17±0.21 | 1.21±0.25 | 0.062 | 0.940 | 0.002 | None |
| | | CT | 1.44±0.84 | 1.18±0.22 | 1.17±0.31 | | | | |
| | LEV | SCT | 1.14±0.34 | 1.18±0.21 | 1.11±0.27 | 0.469 | 0.628 | 0.014 | None |
| | | CT | 1.26±0.26 | 1.19±0.35 | 1.11±0.19 | | | | |
| Extensor testing Convex/ Concave RMS | UEV | SCT | 1.39±0.73 | 1.21±0.36 | 1.14±0.26 | 0.022 | 0.979 | 0.001 | None |
| | | CT | 1.30±0.49 | 1.12±0.32 | 1.09±0.26 | | | | |
| | AV | SCT | 1.54±0.49 | 1.18±0.27 | 1.17±0.43[a] | 0.252 | 0.778 | 0.005 | None |
| | | CT | 1.54±0.70 | 1.25±0.34 | 1.12±0.37 | | | | |
| | LEV | SCT | 1.21±0.35 | 1.13±0.20 | 1.07±0.30 | 0.546 | 0.582 | 0.016 | None |
| | | CT | 1.26±0.45 | 1.18±0.22 | 1.14±0.30 | | | | |
| Flexor strength | | SCT | 11.91±1.81 | 9.97±2.52 | 11.33±2.36 | 0.633 | 0.529 | 0.018 | None |
| | | CT | 11.92±1.85 | 9.77±1.99 | 10.64±2.37 | | | | |
| Extensor strength | | SCT | 16.35±3.13 | 13.37±2.56 | 15.16±2.99 | 2.745 | 0.072 | 0.080 | None |
| | | CT | 17.39±3.03 | 14.12±2.38 | 14.08±2.49 | | | | |
| Flexor endurance | | SCT | 77.83±43.36 | 68.66±36.91 | 83.61±26.22[a] | 1.428 | 0.247 | 0.040 | None |
| | | CT | 76.88±37.35 | 54.66±24.24 | 66.77±22.47 | | | | |
| Extensor endurance | | SCT | 112.00±24.46 | 86.50±31.68 | 121.16±41.33[a] | 18.521 | <0.001 | 0.154 | T0 - T2* T2 - T3** |
| | | CT | 125.72±53.44 | 83.05±23.24 | 93.77±33.08 | 11.869 | <0.001 | | T0-T2** T0-T3** |

T0: preoperative; T2: 3-month postoperative; T3: 6-month postoperative; *: $P<0.05$; **: $P<0.01$; a: compared with CT, $P<0.05$.

Within-group comparisons at discharge revealed both groups exhibited decreased pain and function domain scores versus preoperative values ($P<0.01$), with no significant differences in mental health domains ($P>0.05$). At the 3-month postoperative assessment: in the SCT group, no significant differences versus preoperative baselines were detected for pain, function, or mental health scores ($P>0.05$), but self-image scores were superior to preoperative values ($P<0.01$); in the CT group, no significant differences were found for pain or mental health scores ($P>0.05$), function scores were inferior to preoperative baselines ($P<0.01$).. At the 6-month postoperative assessment: in the SCT group, no significant differences were observed for pain, function, or mental health scores versus preoperative values ($P>0.05$); in the CT group, no significant differences were detected for pain, mental health, or self-image scores ($P>0.05$), but function scores remained inferior to preoperative values ($P<0.05$). Complete results are presented in Table 7 and Fig 9.

**Table 4. Comparative analysis of convex/concave RMS outcomes in paraspinal muscles between the two patient cohorts across three testing maneuvers.**

| Category | | Preoperative | | T | P | 3-month postoperative | | T | P | 6-month postoperative | | T | P |
|---|---|---|---|---|---|---|---|---|---|---|---|---|---|
| | | SCT | CT | | | SCT | CT | | | SCT | CT | | |
| Static standing Convex/concave RMS | UEV | 1.33±0.31 | 1.25±0.34 | −0.024 | 0.472 | 1.20±0.33 | 1.19±0.26 | 0.147 | 0.728 | 1.15±0.41 | 1.18±0.34 | −0.557 | 0.842 |
| | AV | 1.78±0.53 | 1.61±0.49 | 0.908 | 0.344 | 1.20±0.26[b] | 1.18±0.34[b] | 0.133 | 0.751 | 1.19±0.31[b] | 1.20±0.35[b] | −0.356 | 0.945 |
| | LEV | 1.48±0.40 | 1.35±0.32 | 1.298 | 0.323 | 1.15±0.28[a] | 1.18±0.16 | 0.137 | 0.552 | 1.13±0.39[a] | 1.14±0.29[a] | 0.776 | 0.962 |
| Flexor testing Convex/concave RMS | UEV | 1.48±0.39 | 1.33±0.32 | 0.100 | 0.203 | 1.13±0.27[b] | 1.12±0.17[a] | −0.313 | 0.892 | 1.21±0.33[a] | 1.14±0.20 | −0.205 | 0.443 |
| | AV | 1.56±0.73 | 1.44±0.84 | 1.097 | 0.662 | 1.17±0.21[a] | 1.18±0.22[a] | −0.287 | 0.898 | 1.21±0.25[a] | 1.17±0.31 | −0.080 | 0.737 |
| | LEV | 1.14±0.34 | 1.26±0.26 | 0.441 | 0.237 | 1.18±0.21 | 1.19±0.35 | −0.130 | 0.856 | 1.11±0.27 | 1.11±0.19[a] | 0.339 | 0.983 |
| Extensor testing Convex/concave RMS | UEV | 1.39±0.73 | 1.30±0.49 | −0.014 | 0.681 | 1.21±0.36 | 1.12±0.32 | −0.105 | 0.426 | 1.14±0.26 | 1.09±0.26[a] | −0.827 | 0.544 |
| | AV | 1.54±0.49 | 1.54±0.70 | −0.555 | 0.970 | 1.18±0.27[a] | 1.25±0.34[a] | 0.232 | 0.473 | 1.17±0.43[a] | 1.12±0.37[a] | −0.927 | 0.737 |
| | LEV | 1.21±0.35 | 1.26±0.45 | −1.204 | 0.748 | 1.13±0.20 | 1.18±0.22 | −0.183 | 0.437 | 1.07±0.30 | 1.14±0.30 | −0.022 | 0.473 |

a: significantly different from T0 (P<0.05); b: significantly different from T0 (P<0.01).

**Table 5. Comparative analysis of trunk muscular strength and endurance between the two patient cohorts.**

| Category | Preoperative | | T/F | P | 3-month postoperative | | T/F | P | 6-month postoperative | | T/F | P |
|---|---|---|---|---|---|---|---|---|---|---|---|---|
| | SCT | CT | | | SCT | CT | | | SCT | CT | | |
| Flexor strength (kg) | 11.91±1.81 | 11.92±1.85 | −0.018 | 0.986 | 9.97±2.52[b] | 9.77±1.99[b] | 0.260 | 0.796 | 11.33±2.36 | 10.64±2.37[b] | 0.961 | 0.388 |
| Extensor strength (kg) | 16.50±3.01 | 17.39±3.03 | −0.884 | 0.383 | 13.37±2.56[b] | 14.12±2.38[b] | −0.901 | 0.374 | 15.16±2.99 | 14.08±2.49[b] | 1.172 | 0.249 |
| Flexor endurance (N) | 77.83±43.36 | 76.88±37.35 | 0.070 | 0.945 | 68.66±36.91 | 54.66±24.24[b] | 1.345 | 0.188 | 83.61±26.22 | 66.77±22.47 | 2.068 | 0.046* |
| Extensor endurance (N) | 112.00±24.46 | 125.72±53.44 | 0.981 | 0.329 | 86.50±31.68[b] | 83.05±23.24[b] | 0.138 | 0.712 | 121.16±41.33 | 93.77±33.08[b] | 4.817 | 0.035* |

Note: *: *P*<0.05; **: *P*<0.01; a: significantly different from T0 (P<0.05); b: significantly different from T0 (P<0.01).

## Discussion

In this single-blind RCT, the addition of Schroth 3D scoliosis-specific corrective exercises to core stabilization training was significantly more effective than core training alone in promoting pelvic balance, increasing trunk extensor muscular endurance, and further improving self-image scores at six months postoperatively. Despite these functional benefits, the combined intervention did not produce significant improvements in paraspinal muscular balance or other radiographic parameters, such as spontaneous minor-curve correction rates. Overall, these findings provide robust evidence supporting the integration of Schroth-specific exercises into postoperative rehabilitation protocols for adolescents with idiopathic scoliosis, particularly to optimize functional recovery and self-perception.

Established theories propose two critical phases in AIS progression: initial deformity arising from neuromuscular system deficits, followed by balance deterioration secondary to deformity-induced neural dysfunction [29]. Surgical intervention corrects spinal deformity and preoperative coronal imbalance; however, literature reports postoperative coronal imbalance incidence persists at 12.5%−50%, adversely affecting patient appearance and surgical satisfaction [30,31]. Gao et al. [32] observed improved coronal balance in non-surgical AIS patients following a 2-year Schroth training program. The absence

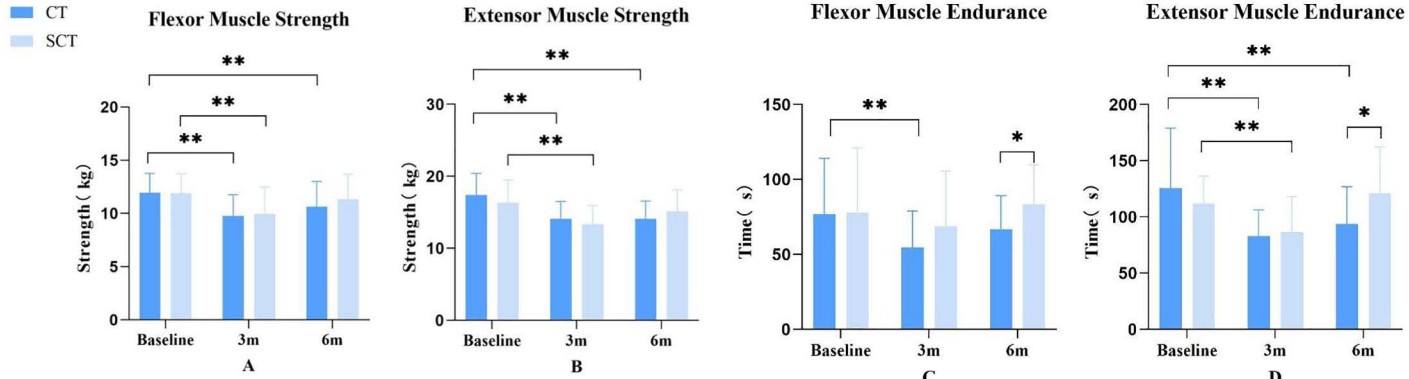

**Fig 8. Changes in trunk muscular strength and endurance in the two groups at preoperative, 3-month postoperative, and 6-month postoperative timepoints.** **(A)** Trunk flexor strength. **(B)** Trunk extensor strength. **(C)** Trunk flexor endurance. **(D)** Trunk extensor endurance. *: $P<0.05$; **: $P<0.01$.

**Table 6. Comparative analysis of SRS-22 outcomes within the groups.**

| Outcomes | | T0 Mean±SD | T1 Mean±SD | T2 Mean±SD | T3 Mean±SD | Time*Group F | P | Partial η² | Pairwise comparison |
|---|---|---|---|---|---|---|---|---|---|
| Pain | SCT | 4.41±0.60 | 3.88±0.65 | 4.31±0.52 | 4.40±0.50 | 0.795 | 0.481 | 0.023 | None |
| | CT | 4.37±0.52 | 3.98±0.60 | 4.14±0.54 | 4.28±0.48 | | | | |
| Function | SCT | 4.27±0.80 | 3.38±0.84 | 3.94±0.48 | 4.23±0.50 | 0.948 | 0.421 | 0.027 | None |
| | CT | 4.36±0.47 | 3.60±0.66 | 3.91±0.49 | 4.05±0.48 | | | | |
| Self-image | SCT | 3.35±0.82 | 3.65±0.46 | 4.02±0.49 | 4.24±0.49[a] | 8.902 | <0.001 | 0.108 | T0 - T2**<br>T0 - T3**<br>T1 - T3** |
| | CT | 3.57±0.45 | 3.67±0.61 | 3.87±0.58 | 3.77±0.79 | 1.704 | 0.186 | | None |
| Mental Health | SCT | 3.91±0.47 | 3.78±0.47 | 3.91±0.59 | 4.08±0.66 | 0.672 | 0.545 | 0.019 | None |
| | CT | 4.05±0.48 | 3.92±0.61 | 4.00±0.64 | 4.03±0.77 | | | | |

T0: preoperative; T1: discharge; T2: 3-month postoperative; T3: 6-month postoperative; *: $P<0.05$; **: $P<0.01$; a: compared with CT, $P<0.05$.

of a statistically significant improvement in coronal balance within the SCT group could potentially be due to the limited 6-month follow-up period. Similarly, the absence of a statistically significant inter-group difference in the correction of secondary curves at the 6-month follow-up may likewise be attributable to the limited intervention and observation period. The modulation of compensatory curves is a long-term adaptive process, and the full therapeutic potential of specific, targeted exercises like Schroth may not be fully captured within a short-term timeframe. This perspective finds support in a case series investigating the ApiFix system, where patients undergoing short-segment thoracic fusion and postoperative Schroth exercises maintained their secondary curves within a range of 22° to 33° over a follow-up period of 6 months to 2 years, without evidence of distal adding-on [14]. This parallel suggests that the stabilization and correction of compensatory curves following intervention is a progressive phenomenon, underscoring that our 6-month findings do not preclude the potential for significant long-term benefits. A study investigating preoperative Scoliosis-Specific Exercise in AIS patients demonstrated that a concise, five-day inpatient program significantly enhanced spinal flexibility in those with rigid curves and facilitated superior postoperative correction [33]. Although the intervention was administered preoperatively,

**Table 7.  Comparative Analysis of SRS-22 Outcomes Between the Two Patient Groups.**

| Category | Preoperative | | T/F | P | Discharge | | T/F | P | 3-month postoperative | | T/F | P | 6-month postoperative | | T/F | P |
|---|---|---|---|---|---|---|---|---|---|---|---|---|---|---|---|---|
| | SCT | CT | | | SCT | CT | | | SCT | CT | | | SCT | CT | | |
| Pain | 4.41±0.60 | 4.37±0.52 | 0.177 | 0.861 | 3.88±0.65b | 3.98±0.60b | −0.475 | 0.638 | 4.31±0.52 | 4.14±0.54 | 0.933 | 0.357 | 4.40±0.50 | 4.28±0.48 | 0.676 | 0.503 |
| Function | 4.27±0.80 | 4.36±0.47 | −0.405 | 0.688 | 3.38±0.84b | 3.60±0.66b | −0.835 | 0.410 | 3.94±0.48 | 3.91±0.49b | 0.205 | 0.839 | 4.23±0.50 | 4.05±0.48a | 1.068 | 0.293 |
| Self-image | 3.35±0.82 | 3.57±0.45 | −1.001 | 0.324 | 3.65±0.46 | 3.67±0.61 | −0.233 | 0.817 | 4.02±0.49b | 3.87±0.58a | 0.799 | 0.430 | 4.24±0.49b | 3.77±0.79 | 4.492 | 0.041* |
| Mental Health | 3.91±0.47 | 4.05±0.48 | −0.898 | 0.376 | 3.78±0.47 | 3.92±0.61 | −0.833 | 0.411 | 3.91±0.59 | 4.00±0.64 | 0.695 | 0.491 | 4.08±0.66 | 4.03±0.77 | 0.231 | 0.819 |

*: *P*<0.05; **: *P*<0.01; a: significantly different from T0 (*P*<0.05); b: significantly different from T0 (*P*<0.01).

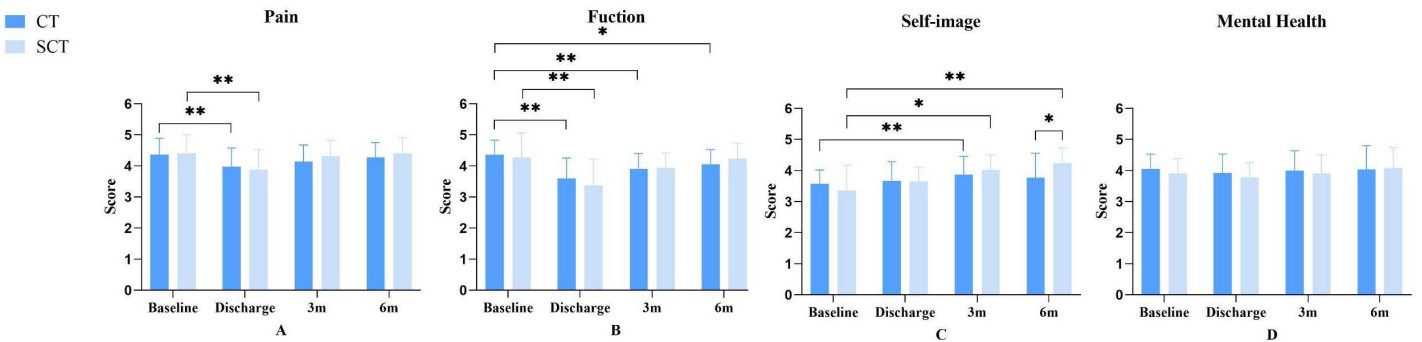

**Fig 9.  SRS-22 Score changes in two groups at preoperative, discharge, 3-Month Postoperative, and 6-Month Postoperative Timepoints.** Figure A shows pain domain scores; Figure B shows mental health domain scores; Figure C shows self-image domain scores; Figure D shows mental health domain scores. *: *P*<0.05; **: *P*<0.01.

this finding underscores the significance of incorporating PSSE into the broader perioperative care pathway to optimize surgical outcomes.

Schroth 3D corrective training incorporates pelvic correction maneuvers combined with breathing techniques that activate muscular contractions in physiologically appropriate patterns. This approach leverages proprioceptive and kinesthetic inputs facilitate optimized restoration of lumbopelvic-hip complex alignment. Aided by the contraction of the concave-side psoas major to derotate and counter-flexion the lumbar curve while restoring lordosis, the lumbar correction was achieved. This maneuver, combined with leveling the pelvis by elevating the convex and depressing the concave iliac crest, led to improved pelvic balance in the SCT group at 6 months post-surgery. Furthermore, Schroth protocols train postoperative patients to consciously maintain corrected postures during their daily activities, thereby improving pelvic balance. Moorthy et al. [34] reported achievable shoulder balance approximately 6 months post-surgery. The absence of intergroup shoulder balance differences in our study may be attributed to several factors, including the selection of upper instrumented vertebra, correction magnitude, and thoracic curve flexibility [35].

Preoperative imbalance in major-curve paraspinal muscles occurs in AIS patients in this study, consistent with established research demonstrating that asymmetric biomechanical characteristics of paraspinal musculature significantly correlate with scoliosis severity [36]. At both 3-month and 6-month postoperative assessments, PMSI values in both groups approached closer to 1 than preoperative values, indicating PSF surgery effectively improves convex/concave imbalance in paraspinal muscles. This aligns with Lu et al.'s findings [11] that paraspinal electromyographic imbalance improves post-fusion but doesn't reach healthy adolescent levels. Chen et al. [37] calculated through tension factors that bilateral paraspinal muscle force equilibrium postoperatively triggers muscular adaptations following spinal biomechanical

realignment. Instrumented fusion reduces scoliotic stress, allowing relaxation of previously overstretched convex-side paraspinal muscles, thereby improving asymmetry. Existing studies confirm core stability training and Schroth therapy improve paraspinal balance in conservatively-treated AIS patients [38]. However, the two rehabilitation protocols in our study demonstrated no differential improvement in postoperative myoelectric symmetry. This equivalence between both groups potentially stems from: 1) Due to the short follow-up period, the predominant surgical impact within the first 6 months may have limited meaningful improvement. 2) slow recovery of surgically disrupted paraspinal muscles, and 3) identical core muscle training administered to both cohorts resulting in comparable convex/concave balance restoration.

PSF procedures using electrocautery for paraspinal muscle separation damage paraspinal musculature, segmental vasculature, and dorsal rami of spinal nerves. This damage accelerates muscular atrophy, degeneration, and necrosis, ultimately inducing functional impairment [39]. As key core stabilizers, abdominal muscle groups experience no direct surgical disruption and consequently maintain structural and functional integrity; however, postoperative pain restricts early postoperative mobilization. Consequently, both groups exhibited diminished flexor and extensor muscular strength and endurance at 3 months postoperatively, confirming PSF induced core musculature dysfunction persisted without recovery. To ensure intervention safety and prevent instrumentation loosening from excessive mechanical loading during early rehabilitation, the SCT cohort incorporated only rotational angular breathing and posture management during the initial three-month training period. These low-efficacy interventions failed to elicit differential strength or endurance improvements compared to the CT cohort. During postoperative months 4–6, the Schroth group was supplemented with 3D corrective maneuvers requiring sustained isometric contractions of trunk stabilizers. These exercises generate longitudinal spinal extension forces, thereby enhancing overall trunk muscular functionality. This targeted approach may explain why, after 6 months of intervention, the SCT cohort demonstrated superior improvement in spinal extensor muscular endurance compared to the CT cohort. Although spared from direct surgical trauma, the trunk flexors as part of the core musculature alongside the extensors are hypothesized to affect functional recovery and balance. Consequently, the finding of greater trunk flexor endurance in the SCT group at 6 months, while notable, must be viewed as preliminary and exploratory given the small sample size.

The SCT cohort demonstrated significantly superior self-image domain scores compared to the CT cohort. This advantage stems from the Schroth protocol's emphasis on training postoperative AIS patients to consciously maintain corrected postures during activities through postural correction and maintenance. yielding greater self-image improvement than core training alone. For pain and functional recovery, the short-term postoperative functional impairment observed in our study aligns with existing literature [9,10]. Notably, while both cohorts restored pain scores to preoperative levels by the 3-month follow-up, the SCT cohort achieved functional recovery equivalent to preoperative baselines, whereas the CT cohort exhibited persistent functional deficits. The Schroth training enhanced postoperative trunk muscular function and mobility, facilitating superior reintegration into daily activities. However, no intergroup differences emerged in mental health scores, attributable to the lack of psychological guidance or support in both rehabilitation protocols. Postoperative psychological health issues in AIS patients warrant attention; future rehabilitation programs should prioritize psychological impact assessment and supportive interventions.

## Limitations and future research

This study has limitation that warrant further consideration. the postoperative follow-up period was limited to six months, which is relatively short for assessing the long-term effects of the intervention. Since the first six months post-surgery are predominantly influenced by surgical factors, some parameters such as spontaneous minor curve correction rates showed no significant differences between groups. Therefore, longer-term follow-up is necessary to fully evaluate the sustained efficacy of Schroth exercises in surgically treated AIS patients. Due to the single-center nature and modest sample size, the generalizability of this study is limited. Furthermore, the limited statistical power for secondary outcomes precluded multiple-comparison corrections, rendering these results exploratory. Consequently, larger, multi-center studies are

required to validate these observations. The SRS-30 questionnaire [40] offers a distinct advantage over the SRS-22 for postoperative assessment by specifically assessing patient satisfaction with truncal and shoulder balance, which serves as a direct proxy for key surgical success. Consequently, we identify the use of the SRS-22 as a study limitation and advocate for the adoption of the SRS-30 in future research to more precisely evaluate patient-centered surgical outcomes.

## Conclusion

The present study provides preliminary evidence that a 6-month program of Schroth exercises combined with core training offers greater benefits in pelvic balance, trunk extensor muscle endurance recovery and enhancement of self-image compared to core training alone in patients following selective thoracic fusion. Future research is needed to further explore the role and long-term effects of Schroth exercises in AIS patients undergoing fusion at different spinal segments, particularly regarding its influence on the maintenance of correction in non-fused secondary curves.

## Author contributions

**Conceptualization:** Fanyuan Meng, Kerong Li, Rui Yang, Lijuan Ao.

**Data curation:** Fanyuan Meng.

**Formal analysis:** Fanyuan Meng.

**Investigation:** Kerong Li.

**Methodology:** Zhi Zhao.

**Project administration:** Wei Wang, Rui Yang, Zhi Zhao.

**Resources:** Wei Wang.

**Supervision:** Cong Wang.

**Validation:** Cong Wang, Lijuan Ao.

**Visualization:** Moxian Chen.

**Writing – original draft:** Kerong Li, Moxian Chen.

**Writing – review & editing:** Moxian Chen.

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
