## [Decision Letter · Decision Letter 0]

31 Oct 2025

Dear Dr. Meng,

Thank you for submitting your manuscript to PLOS ONE. After careful consideration, we feel that it has merit but does not fully meet PLOS ONE’s publication criteria as it currently stands. Therefore, we invite you to submit a revised version of the manuscript that addresses the points raised during the review process.

We look forward to receiving your revised manuscript.

Kind regards,

Holakoo Mohsenifar

Academic Editor

PLOS ONE

Journal Requirements:

“This work was supported by the Yunnan Provincial Department of Science and Technology-Kunming Medical University Joint Special Project for Applied Basic Research (202201AY070001-014), 2024 Yunnan Provincial University Science and Technology Project for Serving Key Industries (FWCY-BSPY2024074), Natural Science Foundation of Guangdong Province (2022A1515010169).”

“This work was supported by the Yunnan Provincial Department of Science and Technology-Kunming Medical University Joint Special Project for Applied Basic Research (202201AY070001-014), 2024 Yunnan Provincial University Science and Technology Project for Serving Key Industries (FWCY-BSPY2024074), Natural Science Foundation of Guangdong Province (2022A1515010169).”

4. In the online submission form you indicate that your data is not available for proprietary reasons and have provided a contact point for accessing this data. Please note that your current contact point is a co-author on this manuscript. According to our Data Policy, the contact point must not be an author on the manuscript and must be an institutional contact, ideally not an individual. Please revise your data statement to a non-author institutional point of contact, such as a data access or ethics committee, and send this to us via return email. Please also include contact information for the third party organization, and please include the full citation of where the data can be found.

6. Please be informed that funding information should not appear in the Acknowledgments section or other areas of your manuscript. We will only publish funding information present in the Funding Statement section of the online submission form. Please remove any funding-related text from the manuscript.

Reviewers' comments:

Reviewer's Responses to Questions

**Comments to the Author**

1. Is the manuscript technically sound, and do the data support the conclusions?

Reviewer #1: Yes

Reviewer #2: Partly

Reviewer #3: Yes

2. Has the statistical analysis been performed appropriately and rigorously?

Reviewer #1: No

Reviewer #2: No

Reviewer #3: Yes

3. Have the authors made all data underlying the findings in their manuscript fully available?

Reviewer #1: Yes

Reviewer #2: Yes

Reviewer #3: Yes

4. Is the manuscript presented in an intelligible fashion and written in standard English?

Reviewer #1: Yes

Reviewer #2: Yes

Reviewer #3: Yes

Reviewer #1: Dear Editor,

Thank you for your kind invitation to review the manuscript entitled “Comparative Efficacy of Schroth and Core Training on Optimizing Trunk Balance and Early Function in Postoperative Adolescent Idiopathic Scoliosis: A Single-Blind Randomized Controlled Trial.”

In this study, the authors aimed to investigate the effectiveness of adding Schroth exercises to core stabilization exercises in patients with AIS who underwent surgical treatment.

Rehabilitation studies conducted after scoliosis surgery are quite limited in the literature. Therefore, the authors have undertaken an important study that may fill a gap in the existing literature.

My detailed comments regarding the manuscript are presented below.

Title and Abstract

The title reflects the overall content of the study; however, it may be a bit long for readers. The authors may consider shortening it slightly.

In this section, I recommend not using “S” for scoliosis classification. Since Schroth exercises were applied, you may use the Schroth classification, or as you focused on postoperative rehabilitation, the Lenke classification would be more appropriate.

As the study involves postoperative evaluation, the SRS-30 questionnaire, which includes postoperative assessment items, could have been used. However, since the study has already been completed, this cannot be changed.

Please report some p-values in the Results section as actual numerical values.

Choose keywords according to MeSH terminology.

Introduction

Pay attention to the use of references throughout the manuscript and ensure that appropriate references are cited. For example, a validation study has been cited for the definition of scoliosis (Ref 1).

Add a reference for Lines 78–79.

The Introduction could be shortened and simplified. I suggest deleting sentences that interrupt the logical flow.

Add the study hypothesis at the end of this section.

Method

In the inclusion criteria, avoid using the “S-type” scoliosis classification. Instead, select a classification method appropriate to your study and describe it in the Methods section.

Provide references for both the core and Schroth exercises.

Clarify where the exercises were performed — were they part of a home program or supervised sessions by a physiotherapist? How many supervised sessions were conducted?

How many sessions were required for the children to learn the exercises?

Were all exercises taught by the same physiotherapist?

If the cases were blinded to group assignment, how was informed consent obtained? Please explain.

Was the rehabilitation protocol initiated on postoperative day 1?

How was exercise progression ensured? Please explain for the readers.

Add references for all assessment methods, including the Cobb method. Indicate that the 2nd, 3rd, and 4th measurements were taken from radiographs.

Provide references for the EMG evaluations and for PMSI.

As a suggestion, the SRS-30 questionnaire could have been used for postoperative evaluation.

Statistical Analysis: Since your sample size is below 50, normality could be tested using the Shapiro–Wilk test.

Results

Add the distribution of curve types to Table 1.

Indicate how many vertebrae were fused and whether there was a difference between groups.

Include measurement units in the tables.

Although there are many tables and assessments, it would be beneficial to highlight group differences. Comparing the changes between groups will help determine whether one group is superior to the other.

Discussion

I recommend beginning this section by clearly presenting your primary finding.

The order of presentation of the assessment methods in the Discussion should match the order used in the Methods and Results sections.

Include studies from the literature that focus on postoperative rehabilitation in AIS and discuss your results in relation to these studies.

As a suggestion, since spinal fusion surgery was performed, hanging exercises that disturb the sagittal plane might not have been suitable.

This is because, by nature, AIS patients tend to have hypokyphosis, and in the postoperative period, sagittal plane curvatures are further affected. Hanging exercises may potentially worsen sagittal alignment. However, this is simply a suggestion for your future studies.,

Reviewer #2: This was a simple single-blinded randomized controlled trial enrolling 46 AIS patients. The analyses intended appear to be reasonable. However, although justified based on a single rather large effect size of 1.02 using the G-Power software, the sample size is actually inadequate for the number of endpoints, time assessments and interactions. The investigators have to better justify this one effect size being relevant to all the objectives (The protocol section 3.1 of the appendix is not very helpful). If this is a repeated measures design, then in the results , one should see the results for each endpoint first giving the overall treatment effect, time effect and interaction effect before doing separate analyses at each of the time points. The many comparisons being made reads like an exploratory analysis.

Also the adjustment made for multiple comparisons is not evident in the results section when everything is compared to p=0.01, 0.05 or 0.001. Please be sure that all adjustments to the type I error are , in fact, valid. For example in Table 4 the p-value for flexor endurance is not significant at 0.05 as the adjusted p-value for the four tests done in this table is 0.0125.

Also, specifically, what is the related samples non parametric test being used for the repeated measures ANOVA, if needed? Also, in the limitation section of the paper the reader should be cautioned of the small sample size from a single institution and that the p-values should be interpreted with caution as should the generalization of the results.

Reviewer #3: This study interested me because, based on my clinical experience, minor lumbar curvature can worsen and frontal balance may be affected after thoracic posterior spinal fusion (PSF) surgery, potentially necessitating secondary surgery.

Schroth-based exercises (PSSE), possibly combined with a Chêneau brace, may help prevent the need for additional lumbar fusion.

While the study's design and outcome measures were well presented, I hoped for more focus on lumbar curve changes. A longer follow-up might have provided better insights, as no significant changes were reported after six months.

I believe assessing extensor muscle endurance is crucial for evaluating patient function, while measuring flexor strength may be less important given the potential for paraspinal muscle damage post-PSF. Additionally, targeting the right Iliopsoas muscle with specific Schroth exercises could effectively correct left lumbar curvature.

**Do you want your identity to be public for this peer review?** For information about this choice, including consent withdrawal, please see our Privacy Policy

Reviewer #1: No

Reviewer #2: No

Reviewer #3: **Yes: ** Alireza Doroudian

---

## [Author Response · Author response to Decision Letter 1]

9 Dec 2025

Section 1. Response to Editor

Dear Dr. Mohsenifar,

Thank you for your decision and the opportunity to revise our manuscript. We are deeply grateful to you and the reviewers for your careful evaluation, constructive feedback, and insightful suggestions. We truly appreciate the time and effort invested in reviewing our work.

We have taken all comments very seriously and have revised the manuscript

accordingly. Each point raised has been addressed in detail below, and corresponding changes have been made in the manuscript to improve its clarity, rigor, and overall quality. We hope that our responses and revisions meet your expectations. We have also ensured that the revised manuscript adheres fully to the PLOS ONE style and formatting guidelines, as per the provided templates.

Funding: The funding statement was previously provided in the cover letter. The declaration was: "The funders had no role in the study design, data collection and analysis, decision to publish, or preparation of the manuscript." The information was kept out of the main text.

ORCID: I have successfully completed and linked ORCID ID to profile in the submission system.

Data We have revised the Data Availability Statement. Data access requests should now be directed to the Ethics Committee of Kunming Medical University, whose contact email has been provided in the statement.

Below we provide a detailed point-by-point response to all reviewers’ comments. We hope the revised manuscript now meets the expectations of both the reviewers and the journal.

Section 2. Response to Reviewers

Reviewer #1:

Thank you for your kind invitation to review the manuscript entitled “Comparative Efficacy of Schroth and Core Training on Optimizing Trunk Balance and Early Function in Postoperative Adolescent Idiopathic Scoliosis: A Single-Blind Randomized Controlled Trial.”

In this study, the authors aimed to investigate the effectiveness of adding Schroth exercises to core stabilization exercises in patients with AIS who underwent surgical treatment.

Rehabilitation studies conducted after scoliosis surgery are quite limited in the literature. Therefore, the authors have undertaken an important study that may fill a gap in the existing literature.

My detailed comments regarding the manuscript are presented below.

I Title and Abstract

Comment 1: The title reflects the overall content of the study; however, it may be a bit long for readers. The authors may consider shortening it slightly.

Response:

We are grateful to the reviewer for their valuable suggestion. In response, we have revised the manuscript title to the following: "Comparative Efficacy of Schroth and Core Training for Early Postoperative Recovery in Adolescent Idiopathic Scoliosis: A Single Blind Randomized Controlled Trial.

Comment 2: In this section, I recommend not using “S” for scoliosis classification. Since Schroth exercises were applied, you may use the Schroth classification, or as you focused on postoperative rehabilitation, the Lenke classification would be more appropriate.

Response

We thank the reviewer for this insightful suggestion. We agree that using 'S-shaped' and 'C-shaped' was an oversimplification and inappropriate for a surgical cohort. As correctly pointed out, we have revised the text to use the Lenke classification system. Specifically, in the abstract (line 30), we have replaced the description of "S-shaped with a major thoracic curve" with "Lenke type 1" to accurately reflect the patient population.

Comment 3�As the study involves postoperative evaluation, the SRS-30 questionnaire, which includes postoperative assessment items, could have been used. However, since the study has already been completed, this cannot be changed.

Response

We sincerely thank the reviewer for this valuable suggestion. Upon further consideration, we fully agree that the SRS-30 questionnaire, with its additional domains specifically addressing postoperative quality of life, is indeed more suitable for studies focusing on the postoperative period. This insight provides invaluable guidance for our research group's future work in rehabilitation for AIS patients. As the present study has already been completed, we have acknowledged the limitation of not having used the SRS-30 questionnaire in the dedicated section of our manuscript (Lines 514-518).

Comment 4�Please report some p-values in the Results section as actual numerical values.

Response

We are grateful to the reviewer for bringing this to our attention. We have reported some specific statistical values and p-values in the text regarding the statistical results.

Comment 5�Choose keywords according to MeSH terminology.

Response:

We are grateful to the reviewer for this valuable feedback. In response, we have re-selected our keywords based on the MeSH thesaurus. The new keywords: Adolescent, Scoliosis, Spinal Fusion, Exercise Therapy, are all controlled vocabulary terms within MeSH, which indeed improve the discoverability of our manuscript.

II Introduction

Comment 6�Pay attention to the use of references throughout the manuscript and ensure that appropriate references are cited. For example, a validation study has been cited for the definition of scoliosis (Ref 1)

Response

We are grateful to the reviewer for bringing this to our attention. We agree that the previous reference Ref 1 was not appropriate for defining scoliosis and have therefore replaced it with a more suitable citation. The updated reference is: Diarbakerli E, Savvides P, Wihlborg A, Abbott A, Bergström I, Gerdhem P. Bone health in adolescents with idiopathic scoliosis. Bone Joint J. 2020;102-B(2):268–72. All references cited in the manuscript have been carefully examined following the suggestion.

Comment 7: Add a reference for Lines 78–79.

Response:

We thank the reviewer for the suggestion. Reference [3] (2016 SOSORT guidelines) was previously cited at line 85. We have now added an additional citation of reference [3] at lines 87 as recommended.

Comment 8: The Introduction could be shortened and simplified. I suggest deleting sentences that interrupt the logical flow.

Response:

We are grateful to the reviewer for this valuable feedback. In response, we have thoroughly revised the Introduction to enhance its conciseness and logical coherence. We have deleted several sentences that were repetitive or disrupted the narrative flow, with the majority of changes concentrated in the first and third paragraphs.

Comment 9:Add the study hypothesis at the end of this section.

Response:

We are grateful to the reviewer for this insightful recommendation. In response, we have formulated and stated the primary research hypothesis, which aligns directly with our main objective. It is hypothesized that patients undergoing the combined Schroth and core training intervention will exhibit significantly better preservation of spinal alignment and postural correction at the 6-month assessment compared to those in the active control group performing core training only. (Lines 137-140).

III Method

Comment 10: In the inclusion criteria, avoid using the “S-type” scoliosis classification. Instead, select a classification method appropriate to your study and describe it in the Methods section.

Response

We would like to express my sincere thanks to the reviewer for this constructive suggestion. Based on this comment, the previous description of 'S-shaped curve with a major thoracic curve' in the inclusion criteria has been replaced with the standard 'Lenke type 1' classification. We agree that the Lenke classification provides a more accurate and clinically relevant definition for patients undergoing surgical intervention.

Comment 11�Provide references for both the core and Schroth exercises.

Response:

We thank the reviewer for this valuable comment. We have now revised the manuscript to include relevant references on both Schroth therapy and core exercise to better support our statements.

Comment 12: Clarify where the exercises were performed — were they part of a home program or supervised sessions by a physiotherapist? How many supervised sessions were conducted?

How many sessions were required for the children to learn the exercises?

Were all exercises taught by the same physiotherapist?

Was the rehabilitation protocol initiated on postoperative day 1?

How was exercise progression ensured? Please explain for the readers.

Response:

We thank the reviewer for this important question. We have provided a clearer description of the intervention protocol.

First, given that all participants were school-aged children, the 6-month intervention program could not be conducted entirely in a hospital setting under direct therapist supervision. Therefore, the program consisted of a combination of therapist-supervised sessions and home-based exercises. Specifically, patients in both groups attended one weekly supervised session at the clinic with the same therapist. For the remainder of the week, they performed an additional 2-3 home-based sessions per week, which were overseen by their parents.

Second, initial training involving 4-5 sessions conducted by the same certified Schroth PT, who ensured participants and their caregivers could accurately perform the exercises at home or school.

Third, all exercise sessions were instructed by the same therapist certified in the International Schroth Scoliosis Therapy (ISST) method. This approach was taken to mitigate the potential heterogeneity in intervention effects that could arise from differences in therapeutic skill and delivery.

Fourth, reflecting our commitment to the ERAS protocol, rehabilitation began on postoperative day 1 for both study groups. A key safety protocol required the therapist to deliver all interventions in collaboration with and under the guidance of the attending orthopedic surgeon. The initial regimen included simple breathing exercises, bed-mounted exercises for the lower limbs, facilitated ambulation, and posture control exercises utilizing mirror visual feedback.

At last, this hybrid model, with its weekly clinic visits, allowed the therapist to closely monitor each patient's performance, progressively modify the exercise regimen, and ensure the principle of gradual progression was followed throughout the intervention.

We have added a detailed description of the intervention protocol to the manuscript to provide greater clarity for readers regarding the rehabilitation procedures. (Lines 185-194)

Comment 13�If the cases were blinded to group assignment, how was informed consent obtained? Please explain.

Response

We appreciate the reviewer's comment on this critical methodological detail. To clarify, the informed consent process was completed before group assignment took place. During this process, all adolescent participants and their parents/legal guardians received comprehensive information about the study. This included an explanation that the study was comparing two different physical therapy approaches after surgery, and that they would be randomly allocated to one of them without being told which one. The key elements of both rehabilitation programs were described in general terms (e.g., "a program focusing on core muscles" vs. "a program integrating core training with specific scoliosis exercises"), without disclosing the specific hypothesis about which was expected to be superior. This method is aligned with CONSORT guidelines for single-blind trials and was approved by our institutional ethics committee, ensuring that blinding was maintained without compromising the ethical validity of the consent. I have already added this passage to Trial design and Randomization and Blinding.

Comment 14�Add references for all assessment methods, including the Cobb method. Indicate that the 2nd, 3rd, and 4th measurements were taken from radiographs.

Response:

We are grateful to the reviewer for raising these important points. We have now supplemented the manuscript with key references pertaining to pelvic balance. Additionally, as suggested, we have provided a clearer description of the radiographic measurement process by adding the following statement: "At each predefined assessment timepoint, radiographic parameters (including Cobb angles) were measured from standing radiographs using RadiAnt DICOM Viewer software." (Lines 214-216)

Comment 15: Provide references for the EMG evaluations and for PMSI.

Response:

We are grateful to the reviewer for this valuable feedback. In response, we have incorporated appropriate citations into the manuscript to substantiate the methodologies pertaining to EMG evaluations and the PMSI.

Comment 16: As a suggestion, the SRS-30 questionnaire could have been used for postoperative evaluation.

Response:

We are grateful to the reviewer for this insightful recommendation. Our team has reviewed the SRS-30 and recognizes its superior applicability for postoperative research, particularly given the supplementary items related to postoperative quality of life that are not captured by the SRS-22. This perspective is immensely helpful for guiding our subsequent investigations into post-surgical rehabilitation for AIS. Given that the current study is finalized, we will explicitly state the omission of the SRS-30 as a study limitation in the revised manuscript.

Comment 17: Statistical Analysis: Since your sample size is below 50, normality could be tested using the Shapiro–Wilk test.

Response:

We appreciate the valuable suggestion from the reviewer. Given that the sample size of this study is less than 50, we have supplemented the Shapiro–Wilk test to verify the normality of the data.

IV Results

Comment 18�Add the distribution of curve types to Table 1. Indicate how many vertebrae were fused and whether there was a difference between groups.

Respone

We thank the reviewer for these critical suggestions regarding the baseline characteristics.

Baseline Table (Table 1): As recommended, we have added the distribution of curve types (Lenke classification) and the number of fused vertebrae to Table 1. Statistical analysis confirmed that there were no significant differences between the two groups in these parameters, reinforcing the comparability of the groups at baseline.

In accordance with the reviewer's suggestion, we employed the Lenke classification. However, due to the limited sample size, patients were categorized into the three primary lumbar modifier subtypes (Lenke 1A, 1B, and 1C), and a further subdivision based on sagittal thoracic modifiers�hypokyphosis, normal, hyperkyphosis�was not performed.

Comment 19� Include measurement units in the tables.

Respone

We are grateful to the reviewer for their keen observation. In response, we have conducted a systematic check of all tables and have supplemented the units of measurement accordingly to ensure full clarity.

Comment 20�Although there are many tables and assessments, it would be beneficial to highlight group differences. Comparing the changes between groups will help determine whether one group is superior to the other.

Response

We are grateful to the reviewer for this valuable feedback. In response, we have thoroughly revised the presentation of the results in our manuscript. The changes, now visible in track-change mode, are aimed at more effectively highlighting the comparative outcomes between the two intervention groups and explicitly stating where one approach proved superior.

V Discussion

Comment 21�I recommend beginning this section by clearly presenting your primary finding.

Response

We are grateful to the reviewer for this valuable insight. We fully agree that the discussion should prioritize the study's primary findings. In response, we have reframed the opening of the discussion section, removing the paragraph on muscle and self-image to place greater emphasis on comparing the outcomes between the two groups.

Comment 22�The order of presentation of the assessment methods in the Discussion should match the order used in the Methods and Results sections.

Response:

We are grateful to the reviewer for this valuable feedback on improving the manuscr

---

## [Decision Letter · Decision Letter 1]

23 Dec 2025

Comparative Efficacy of Schroth and Core Training for Early Postoperative Recovery in Adolescent Idiopathic Scoliosis: A Single Blind Randomized Controlled Trial

PONE-D-25-51249R1

Dear Dr. Fanyuan Meng,

We’re pleased to inform you that your manuscript has been judged scientifically suitable for publication and will be formally accepted for publication once it meets all outstanding technical requirements.

Kind regards,

Holakoo Mohsenifar

Academic Editor

PLOS One

Additional Editor Comments (optional):

Reviewers' comments:

Reviewer's Responses to Questions

**Comments to the Author**

Reviewer #2: All comments have been addressed

2. Is the manuscript technically sound, and do the data support the conclusions?

Reviewer #2: Yes

3. Has the statistical analysis been performed appropriately and rigorously?

Reviewer #2: Yes

4. Have the authors made all data underlying the findings in their manuscript fully available?

Reviewer #2: Yes

5. Is the manuscript presented in an intelligible fashion and written in standard English?

Reviewer #2: Yes

Reviewer #2: All comments have been addressed and the revisions incorporated into the paper.

XXXXXXXXXXXXXXXXXXXXXXXXXXXXXXXXXXXXXXXXXXXXXXXXXXXXXXXXXXXXXXXXXXXXXXXXXXXXXXXXXXXXXXXXXXXXXXXXXXXXXXXXXXXXXXXXXXXXXXXXXXXX

**Do you want your identity to be public for this peer review?** For information about this choice, including consent withdrawal, please see our Privacy Policy

Reviewer #2: No

---

## [Editor Report · Acceptance letter]

PONE-D-25-51249R1

PLOS One

Dear Dr. Meng,

I'm pleased to inform you that your manuscript has been deemed suitable for publication in PLOS One. Congratulations! Your manuscript is now being handed over to our production team.

Kind regards,

on behalf of

Dr. Holakoo Mohsenifar

Academic Editor

PLOS One